# First Evidence of Function for *Schistosoma japonicum*
*riok-1* and RIOK-1

**DOI:** 10.3390/pathogens10070862

**Published:** 2021-07-08

**Authors:** Mudassar N. Mughal, Qing Ye, Lu Zhao, Christoph G. Grevelding, Ying Li, Wenda Di, Xin He, Xuesong Li, Robin B. Gasser, Min Hu

**Affiliations:** 1State Key Laboratory of Agricultural Microbiology, College of Veterinary Medicine, Huazhong Agricultural University, Wuhan 430070, China; Mudassar.N.Mughal@vetmed.uni-giessen.de (M.N.M.); Qingye198173@163.com (Q.Y.); bentengzhilu@163.com (L.Z.); lyhappycool@163.com (Y.L.); anniehe1991@gmail.com (X.H.); xuesong.li84@gmail.com (X.L.); 2Biomedical Research Center Seltersberg, Institute of Parasitology, Justus Liebig University Giessen, D-35392 Giessen, Germany; christoph.grevelding@vetmed.uni-giessen.de; 3College of Animal Science and Technology, Guangxi University, Nanning 530005, China; diwenda@gxu.edu.cn; 4Department of Veterinary Biosciences, Faculty of Veterinary and Agricultural Sciences, Melbourne Veterinary School, The University of Melbourne, Parkville, VIC 3010, Australia; robinbg@unimelb.edu.au

**Keywords:** schistosomiasis, *Schistosoma japonicum*, right open reading frame protein kinase (*riok*) genes, *riok*-1, RIOK-1, double-stranded RNA interference (RNAi), chemical inhibition, toyocamycin, developmental and reproductive biology

## Abstract

Protein kinases are known as key molecules that regulate many biological processes in animals. The right open reading frame protein kinase (*riok*) genes are known to be essential regulators in model organisms such as the free-living nematode *Caenorhabditis elegans*. However, very little is known about their function in parasitic trematodes (flukes). In the present study, we characterized the *riok-1* gene (*Sj*-*riok-1*) and the inferred protein (*Sj*-RIOK-1) in the parasitic blood fluke, *Schistosoma japonicum*. We gained a first insight into function of this gene/protein through double-stranded RNA interference (RNAi) and chemical inhibition. RNAi significantly reduced *Sj*-*riok-1* transcription in both female and male worms compared with untreated control worms, and subtle morphological alterations were detected in the ovaries of female worms. Chemical knockdown of *Sj*-RIOK-1 with toyocamycin (a specific RIOK-1 inhibitor/probe) caused a substantial reduction in worm viability and a major accumulation of mature oocytes in the seminal receptacle (female worms), and of spermatozoa in the sperm vesicle (male worms). These phenotypic alterations indicate that the function of *Sj*-*riok-1* is linked to developmental and/or reproductive processes in *S. japonicum*.

## 1. Introduction

In multicellular organisms, protein kinases (PKs) are encoded by large gene families, and regulate cellular processes, including DNA transcription, DNA replication, cell-cycle progression as well as metabolism [1,2]. PKs function to activate/inactivate proteins by catalyzing the transfer of phosphate groups to specific amino acid residues (i.e., Arg, His/Asp and Ser/Thr/Tyr) on their target proteins and, thus, play a regulatory role in many cell signaling pathways [1]. PKs can be classified as eukaryotic protein (ePKs), atypical protein (aPKs) kinases and protein kinase-like (PKL) [3]. For example, of >500 human PKs, <10% are PKL proteins, many of which were known as aPKs. There are 19 families of PKL kinases, one of which is called right open reading-frame kinases (RIOKs) [3].

Multicellular (metazoan) organisms usually have three *riok* genes (called *riok-1*, *riok-2*, and *riok-3*) [4]. However, it has been shown that flatworms (i.e., trematodes and cestodes) lack *riok-3* [5]. Structural information for RIOK proteins in metazoans is limited to the partially-solved crystal structure for RIOK-1 of humans [4], and functional domains of RIOK proteins have been modeled using three-dimensional (3D) structural modeling to assist in predicting and prioritizing kinase inhibitors that might target RIOK-1 of parasitic worms [6,7]. Functional information is available for metazoan model organisms, including the free-living nematode, *C. elegans*, for which investigations have demonstrated that *riok-1* and *riok-2* genes and their products play essential part in the process of ribosome biosynthesis, cell cycle progression, and/or chromosome stability [8,9,10], whereas *riok-3* has received limited attention, likely because it is not an essential gene [11,12,13]. 

There is scant information on the structure and function of *riok-1* and *riok-2* genes of parasitic flatworms, such as representative of the genus *Schistosoma* (blood flukes)—which are dioecious trematodes. Key species include *Schistosoma japonicum*, *S. haematobium*, and *S. mansoni*, which cause the neglected tropical disease (NTD) complex “schistosomiasis”, affecting ~200 million people worldwide [14,15]. In parts of Asia, *S. japonicum* is particularly important because it can be transmitted (via the snail intermediate host) from water buffaloes, cattle, pigs, dogs, or rats to humans [16]. In the human host, infection is initiated by cercariae, free-living larvae released from the snail intermediate host; upon contact with water, cercariae penetrate the skin, transform into schistosomula, enter the blood circulation, and reach the hepatic portal and mesenteric veins, where the adult female and male worms pair up and reproduce. Eggs produced by female worms either penetrate the vessel walls and enter the intestinal lumen, being released via feces into the environment, or are passively transported via blood to the liver and spleen where they become entrapped, inducing the formation of granulomata [15]. 

Patients suffering from schistosomiasis are usually treated with praziquantel (PZQ), as no anti-schistosome vaccine is available. However, due to the widespread and regular use of PZQ in mass treatment programs, there is major concern that schistosomes develop resistance to this compound [17]. Moreover, PZQ fails to affect the juvenile stage of the parasite [18,19] and does not prevent reinfection [17]. These issues motivate efforts to functionally characterize essential genes or their products, particularly those involved in growth, developmental, and reproductive processes in schistosomes [20,21,22], in search for new interventions. In this context, we explored the *riok-1* gene (*Sj-riok-1*) and the function of RIOK-1 (*Sj*-RIOK-1) in *S. japonicum*, and provided the first evidence that this gene is involved in reproductive processes.

## 2. Results

### 2.1. Sj-riok-1 Encodes a Protein with Features Characteristic of RIOK-1

The genomic DNA sequence of *Sj-riok-1* was assembled using data from WormBase (PRJEA34885, scaffold SJC_S002310). The sequence was 6761 bp in length, and it had three exons which were 139, 488, and 642 bp long, respectively, and two introns (3036 and 2456 bp). The coding region of *Sj-riok-1* is 1269 bp long, representing 422 amino acids. A comparison of the inferred amino acid sequence (*Sj*-RIOK-1) with select RIOK-1 orthologs/homologs revealed high sequence identities with those from congeners *S. haematobium* (76.7%) and *S. mansoni* (84.8%), and a lower identity (56.4%) to that from *Clonorchis sinensis*—a carcinogenic liver fluke (Figure 1a). A phylogenetic analysis showed that the sequences of all trematodes formed a clade (with absolute nodal support) to the exclusion of orthologs from other invertebrate and vertebrate species (Figure 1b). 

### 2.2. Transcription in Different Developmental Stages

*Sj-riok-1* was found to be transcribed throughout all developmental stages and sexes assessed (Figure 2a), with transcription being highest in adult females and schistosomules from the lung. Transcript levels of *Sj-riok-1* in single, cultured females for 3 days was 7.2-fold (F_(1,8)_ = 7.578, *p* = 0.0295; t_(6)_ = 3.866, *p* < 0.01) higher than in couples cultured for the same time (Figure 2b). There was a significant reduction of *Sj-riok-1* transcript levels following re-pairing of females and males in vitro (6 days) which had been separated for 3 days prior to re-paring (Figure 2c; t_(6)_ = 2.255, *p* < 0.05 for females; t_(6)_ = 4.910, *p* < 0.001 for males). The results suggest that *Sj-riok-1* is involved in pairing-dependent developmental and/or reproductive processes.

### 2.3. Significant Knockdown of Transcription by RNAi in Both Sexes, and Subtle Morphological Change in the Ovary

As morphological changes are seldomly seen in adult parasitic helminths following short-term RNAi by soaking [23], we elected to assess the specific reduction in gene transcription as the phenotype. Since the *Sj-riok-1* transcript level was highest in the adult stage during the sexual life-phase of *S. japonicum* (Figure 2a), we used paired adult worms for RNAi for a period of 14 days. Results showed that *Sj-riok-1* mRNA transcription was significantly and consistently reduced (Figure 3a) by 75–84% (F_(2,12)_ = 90.24, *p* < 0.0001) in both male and female worms (35- or 28-days old, do) as compared with controls (i.e., worms treated with an irrelevant-dsRNA or without dsRNA) (Figure 3a,b). Although subtle, a significant decrease in ovary size (reduced length and width) was seen in treated worms (*n* = 28) compared with control worms (*n* = 28) (Figure 3c–f).

### 2.4. Toyocamycin Affects Viability and Induces Pathological Changes in the Reproductive Tracts 

Toyocamycin is a competitive, small molecule inhibitor of RIOK1 kinase activity [24]. In vitro treatment of paired adult worms (35- or 28-do) with toyocamycin (1 µM) significantly affected their viability; all worms died after 4–5 days (Figure 4a), whereas untreated controls lived for 12–14 days in culture. Worm pairs started to separate at 24 h, were completely separated at 48 h, and then curled and had a marked swelling in the gut at ≥72 h. At this time point, worm motility, gut peristalsis, and egg production (females) ceased, as compared with untreated controls (Figure 4a–d). Toyocamycin treatment at a lower concentration (0.5 µM) had a similar, but a less intense effect compared with 1 µM (Figure 4a–d). At 1 µM, the effect of toyocamycin on viability (F_(2,24)_ = 246.2, *p* < 0.0001) and egg production (F_(2,44)_ = 71.66, *p* < 0.0001) of 28-do worms (Figure 4c,d) was greater than on older worms (35-do) (viability: F_(2,24)_ = 163.1, *p* < 0.0001; egg production: F_(2,24)_ = 86.42, *p* < 0.0001) (Figure 4a,b). 

Although not detected at 24, 48, or 72 h, some morphological changes were evident in the reproductive tracts at 96 h (Figure 4e). The changes initially detected at 96 h in the reproductive tracts of female and male worms exposed to 1 µM toyocamycin (Figure 4) were investigated in more detail. Toyocamycin treatment of paired adult worms (28- and 35-do) for 96 h led to a cessation of egg-release in female worms (Figure 4b,d). CLSM examination of gonads revealed a significant accumulation of mature oocytes in the oviduct close to the seminal receptacle of females and of spermatozoa in the sperm vesicle of males (28- and 35-do, Figure 4e). An accumulation of oocytes in the uterus was also seen in toyocamycin-treated 35-do females (Figure 4e).

## 3. Discussion

This study showed a significant and (relatively) consistent reduction in *Sj*-*riok-1* transcript levels in both female and male *S. japonicum* using RNAi compared with well-defined control worms (worms treated with dsRNA from an irrelevant-gene, and untreated worms) as well as subtle ovarian alterations. Chemical knockdown of *Sj*-RIOK-1 with toyocamycin—a competitive inhibitor and biochemical probe [24]—led to a substantial reduction in worm viability and pathological changes in the reproductive tracts, including a significant accumulation of mature oocytes in the area of the seminal receptacle that is part of the oviduct in females, and of spermatozoa in the sperm vesicle in males. These alterations suggest that *Sj*-*riok-1* is linked to developmental and/or reproductive processes in *S. japonicum.*

The functional genomic studies revealed that RIOK-1 of *Strongyloides stercoralis* is essential for the development and survival of *S. stercoralis* larvae [25,26], suggesting that RIOK-1 might be an anthelmintic target. Published studies show that RIO kinases homologs are receiving attention as possible targets for the development of anti-cancer treatments and anti-infectives [5,6,7,13,27]. This applies particularly for *riok-1* and *riok-2*, for which genetic manipulation studies have indicated their involvement in fundamental biological mechanisms such as ribosomal biosynthesis, chromosome stability, and/or cell cycle advancement [13,28,29,30,31,32,33,34,35,36,37]. Breugelmans et al. [5] showed that the *riok-1* and *riok-2* genes were present and transcribed in ≥ 52 species of metazoans, including 25 flatworms species (together with trematodes and cestodes). We observed high similarity in sequence and gene structures among RIOK-1 homologs of four trematode and four cestode species (with exon numbers ranging from 3 to 7, and gene lengths ranging from 1618 to 14,971 bp) than reported for nine nematode species [7], which is consistent with previous results [5], indicating relative conservation in *riok* structure in flatworms. For instance, the conserved RIOK-1 signature sequence “S-T-G-K-E-A” in the ATP binding motif is analogous to the signature sequence “G-x-G-K-E-S” of RIOK-2. However, the active site of RIOK-1 “L-V-H-x-D-L-S-E-Y-N” is different to that of “I-H-x-D-o-N-E-F-N” in RIOK-2 [38]. As identified in our alignment, *Sj*-RIOK-1 has Ser165 as part of conserved dipeptide motif present within the flexible loop capable of phosphorylation and autophosphorylation, indicating that *Sj*-RIOK-1 shares common features with the RIOK-1 family. 

This apparent conservation of RIOK-1 for flatworms and distinctiveness from orthologs in mammals and other eukaryote groups (Figure 1b) suggests that this PKL kinase, or elements thereof, may represent a selective target. As most current kinase inhibitors for therapeutic use [39] target nucleotide binding sites, future work could focus on assessing the selectivity of these sites between schistosomes (and other flatworms) and mammalian hosts. However, it needs to be considered that the topologies of these binding domains could be similar in kinases other than RIOK, so that challenges regarding selectivity might arise when designing inhibitors specifically against a pathogen’s RIOK-1. Selectivity not only relates to a discrepancy between pathogen and host RIOK sites, but probably also nucleotide binding domains in any of the many other kinases (*n* = 500) in the kinome of the human host. Therefore, it would be useful to critically appraise the nature and extent of evolutionary diversity in the nucleotide binding domains in RIOKs between flatworms and the principal mammalian host (human). A promising candidate might be the residue at position L289 in the sequence of human RIOK-1, which relates to P197 in *Schistosoma* species (*S. haematobium*). This distinction in the nucleotide-binding domain between these trematodes and the host could be relevant for designing ligands that particularly target RIOK-1 proteins of *S. japonicum* and its congeners, although such work should be assisted by investigating the crystal structures of these flukes. 

As PZQ (a pyrazinoisoquinoline) is used to treat schistosomiasis of humans, this compound is not effective against all developmental stages of schistosomes [40,41]. Therefore, there is an urgency to work toward novel and improved drug targets against flatworms build on deep insights of the molecular biology and development of these worms. In this perspective, it is critical to target gene products that are transcribed or expressed in suitably “druggable” developmental stages of these worms, as is true of RIOKs. *S. japonicum* appears to transcribe *riok-1* in all developmental stages and both sexes, which is similar to findings for *S. haematobium* [42] and *S. mansoni* [20,43], and also other parasitic worms including *Ascaris suum, Brugia malayi*, and *H. contortus* [7]. Such constitutive transcription for *riok-1* in parasitic trematodes and nematodes (studied to date) indicates that RIOK-1 performs essential house-keeping functions pertaining to development and reproduction, which concords with essential roles in ribosome biosynthesis and cell-cycle progression in the well-studied model organisms *C. elegans* and *D. melanogaster* [8,13].

## 4. Materials and Methods

### 4.1. Procurement of the Parasite

The collection, processing, and storage of the different developmental stages (i.e., cercariae, skin-stage and lung-stage schistosomula, juvenile and adult-stage male and female worms, and eggs) of *Schistosoma japonicum* were conducted using established protocols [44].

### 4.2. Cloning of Sj-Riok-1 cDNA from S. japonicum, and Informatic Analyses

Total 3′-end cDNA (using *Sj-riok-1*-specific internal primers Sj-Riok1-F and Sj-Riok1-R, designed from expressed sequence tag (EST) sequence—GenBank accession no. AY810901.1, Appendix A) was prepared as previously described [44]. After cloning and sequencing, the sequence amplified by 3′ RACE-PCR was merged to the known sequence region to generate a full-length *Sj-riok-1* sequence, which was subsequently used in designing two additional primers Sjriok1-ORF-F and Sjriok1-ORF-R (Appendix A) for obtaining the full-length *Sj-riok-1* coding sequence (accession no. MN335243) using the conditions described previously [44]. 

The *Sj*-RIOK-1 protein sequence was inferred and its protein domains, motifs, and/or functional sites inferred using PROSITE [45] and Pfam [46]. RIOK-1 homologs representing 20 species other than *S. japonicum* (Table 1) were extracted from GenBank for sequence alignment with *Sj*-RIOK-1 and phylogenetic analysis employing the neighbor joining (NJ), maximum likelihood (ML), and maximum parsimony (MP) methods in MEGA v.5.0 using the same parameters as previously described [44].

### 4.3. qPCR to Assess Transcript Levels of Different Developmental Stages and from In Vitro Paring Experiment

Quantitative real-time PCR (qPCR) was performed to assess transcript levels in distinct developmental stages of *S. japonicum* as described previously [44] using the primer pair Riok1-qPCR-F/Riok1-qPCR-R for *Sj-riok-1* and the primer pair β-Tubulin-qPCR-F/β-Tubulin-qPCR-R for the *β-tubulin* gene (accession no. AY220457.2) as the reference (Appendix A). The 2^-∆∆Ct^ method [63] was used for relative quantification. The cercarial stage was used as the calibration standard, and data were presented as the mean ± standard deviation.

Adults *S. japonicum* collected from infected mice (42 day infection) were in vitro cultured in the same medium and under the same conditions as described previously [44]. In some experiments, worms were cultured as pairs (*n* = 10) or as individuals (*n* = 10 for each sex) for 3- or 9-day periods. In other experiments, pairs (*n* = 10 each) were cultured for 3 days, separated for 3 days and re-paired for 6 days. qPCR was used to assess *Sj-riok-1* transcription in individual worms or pairs, employing in vitro cultured individual female and male worms from pairs as calibration standard.

### 4.4. Double-Stranded RNA Interference (RNAi)

*Sj-riok-1* (1155 bp, including the RIOK-1 domain) and *egfp* (620 bp) cDNAs were amplified by PCR using primer pair dsRNA-riok1-F/dsRNA-riok1-R and dsRNA-egfp-F/dsRNA-egfp-R (Appendix A), respectively, employing the following conditions: 94 °C for 3 min, then 35 cycles at 94 °C for 40 s, 60 °C (for *Sj-riok-1*) or 62 °C (for *egfp*) for 40 s and 72 °C for 2 min, and final extension step at 72 °C for 10 min. These two cDNAs were each cloned into pMD-19T and their identities confirmed by sequencing. Each of the plasmids was used in the production of dsRNA which was subsequently employed in RNAi using an established soaking method [44]. The transcript levels of dsRNA-treated worms were assessed by qPCR (using untreated females as the calibrator/reference), and worms were microscopically examined for morphological alterations.

### 4.5. Treatment with Toyocamycin

Adult *S. japonicum* (couples) were perfused from mice and maintained in vitro at 37 °C for 96 h in a 5% CO_2_ atmosphere in the culture medium (same as used in the paring experiment) which contains the inhibitor toyocamycin (APExBIO^®^, Houston, TX, USA) dissolved in dimethyl sulfoxide (DMSO) and was refreshed daily. In preliminary experiments, both 28- and 35-do couples were treated with varying concentrations (100, 300, 500 nM, and 1 µM) of toyocamycin in vitro for 96 h (Appendix A). Control groups were incubated with equal volume of DMSO-only under the same conditions. IC50 concentrations and egg count as well as worm viability were determined every 24 h (Appendix A). Worm viability was scored as recommended by WHO-TDR [64] normal motility (3), reduced motility (2), minimal motility (1), and no movement/dead (0).

### 4.6. Microscopic Examination of Worms and Statistical Analyses

Worms were examined by bright field microscopy (BFM) and CLSM as previously described [44]. The length, width, and the area of the ovaries were measured by BFM. Data are representative of the mean ± SD of at least three independent experiments. Statistically significant differences were analyzed by GraphPad Prism v.5.01 (San Diego, CA, USA). A two-way ANOVA with Bonferroni’s multiple comparison test was used for analyzing worm viability and egg production of the different groups including the re-paired worms. One-way ANOVA with multiple comparison of the Tukey-test was used for the other experiments. Values of *p* ≤ 0.05 were considered as statistically significant.

## 5. Conclusions

Our findings represent first evidence for the pairing-dependent influence of *Sj-riok-1* in the development and reproductive maturation of adult female *S. japonicum*. This study contributes towards the understanding of the reproductive processes in *S. japonicum* and suggests RIOK-1 as a potential target as selective anthelmintic therapeutic approach.

## Figures and Tables

**Figure 1 pathogens-10-00862-f001:**
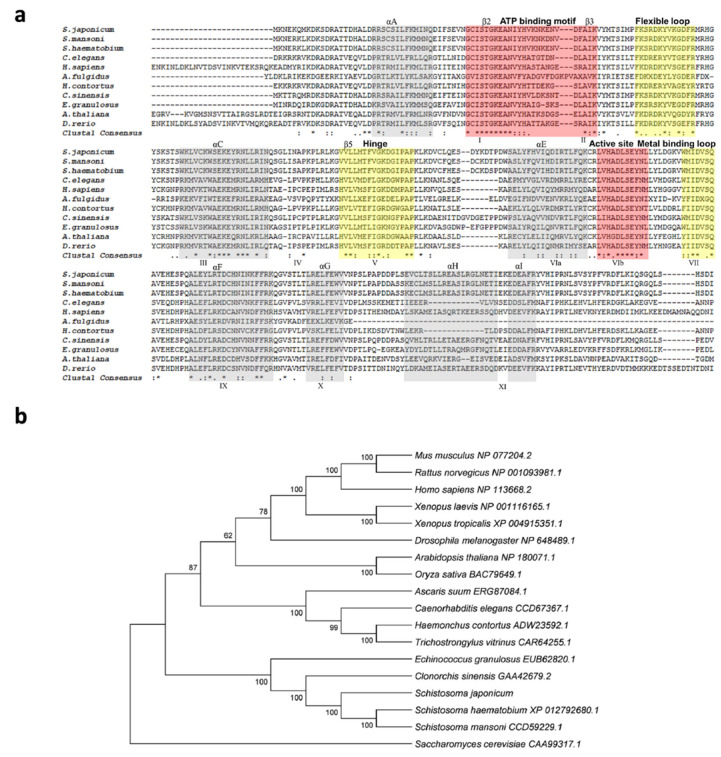
Comparison of *Schistosoma japonicum Sj*-RIOK-1 with RIOK-1s of other organisms. (**a**) Multiple sequence alignment among predicted RIOK-1 proteins from 11 organisms (*Schistosoma japonicum*, *Schistosoma mansoni*, *Schistosoma haematobium*, *Caenorhabditis elegans*, *Homo sapiens*, *Archaeoglobus fulgidus*, *Haemonchus contortus*, *Clonorchis sinensis*, *Echinococcus granulosus*, *Arabidopsis thaliana*, and *Danio rerio*). Alpha helices A-I or beta-sheet structures are colored light gray and labeled above the alignment. The subdomains I-XI are marked below the alignment. Functional domains, including the ATP binding motif and active site (red), flexible loop, hinge, and metal binding loop (yellow), are highlighted and marked above the alignment. Identical residue (*); high similarity (:); limited similarity (.); no similarity (no symbol). (**b**) The neighbor-joining (NJ) tree of RIOK-1 amino acid sequences from a range of organisms. The RIOK-1 of *Saccharomyces cerevisiae* (CAA99317.1) was used as an outgroup, and the bootstrap values are given above or below the branches. GenBank accession numbers are listed besides the species name. Accession numbers and related references of RIOK-1 amino acid sequences used for the multiple alignments and phylogeny are given in Table 1.

**Figure 2 pathogens-10-00862-f002:**
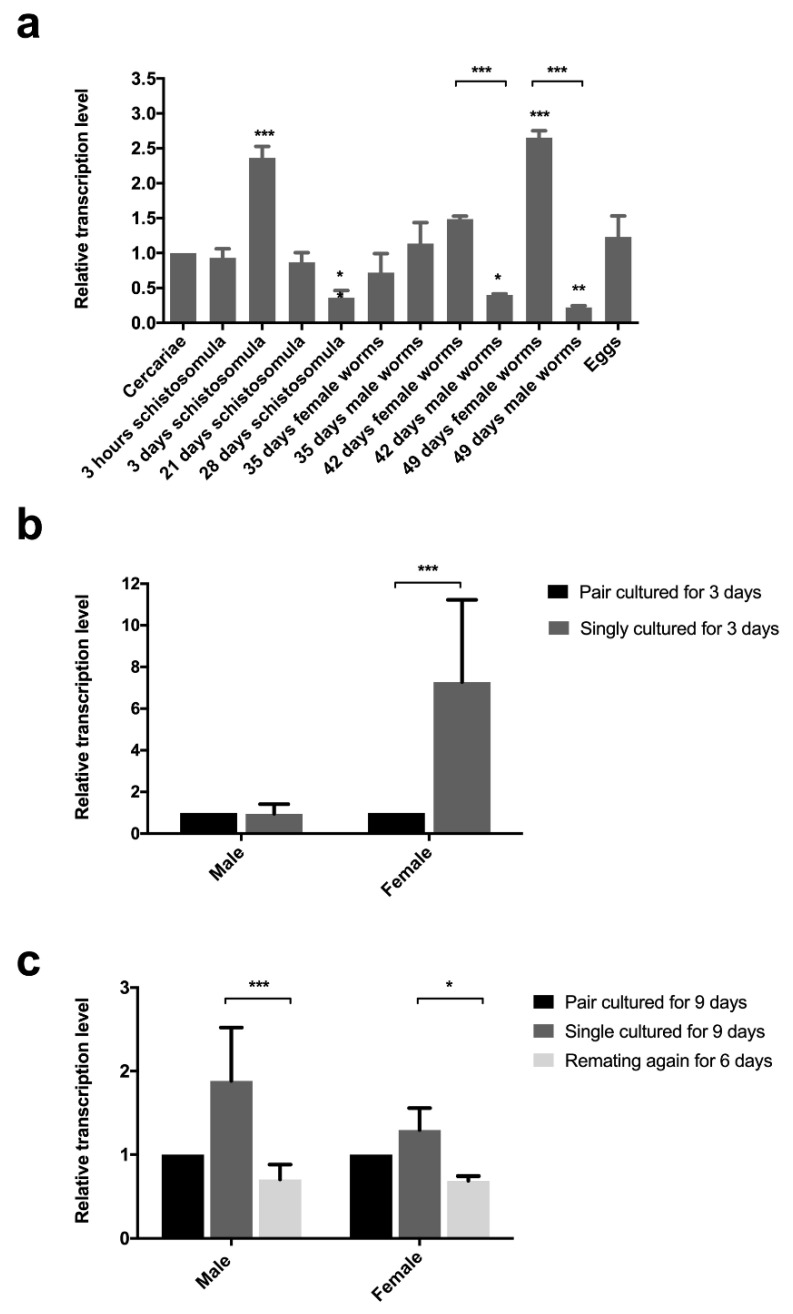
Relative transcription levels of *Sj-riok-1* in various developmental stages, and the effect of pairing on the transcription of this gene in adult males and females of *Schistosoma japonicum*. (**a**) Transcriptional level of *Sj-riok-1* in various developmental stages of *S. japonicum* (female worms from mixed cultures were used and the data normalized relative to the cercarial stage). (**b**,**c**) The effect of pairing on the transcription of *Sj-riok-1* in adult males and females. Data for separated and re-paired worms were normalized relative to the paired worms. Data given are representatives of the mean ± SD of three independent experiments, and statistically significant differences are indicated as * (*p* < 0.05), ** (*p* < 0.01), and *** (*p* < 0.001).

**Figure 3 pathogens-10-00862-f003:**
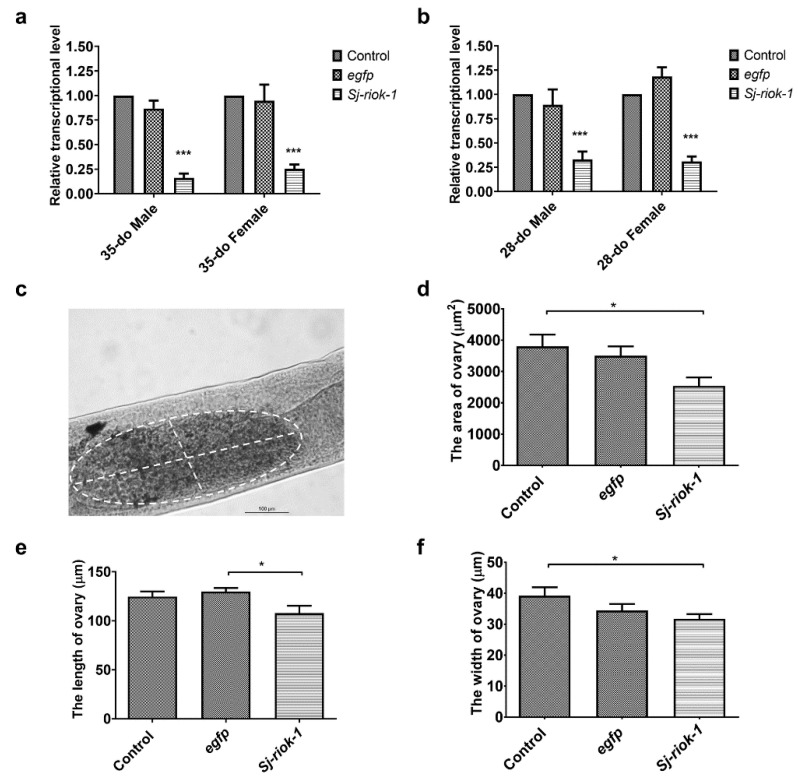
RNAi-mediated knockdown of *Sj-riok-1* gene decreased transcription in 35- or 28-do adult male and female worms and the ovarian dimension of the 35-do adult females of *Schistosoma japonicum*. (**a**,**b**) Relative *Sj-riok-1* transcript levels determined by qRT-PCR in 35- or 28-do worms treated with *Sj-riok-1* dsRNA, non-specific (egfp) dsRNA, and no dsRNA (control). (**c**–**f**) The area, length, and width of the ovary in 35-do females after 2 weeks of treatment with *Sj-riok-1* dsRNA or irrelevant (egfp) dsRNA, including a no-dsRNA (control) were measured using a bright-field microscope. Data are given as the mean ± SD of three independent experiments with 8–10 females in each experiment. * indicates *p* < 0.05; *** indicates *p* < 0.001.

**Figure 4 pathogens-10-00862-f004:**
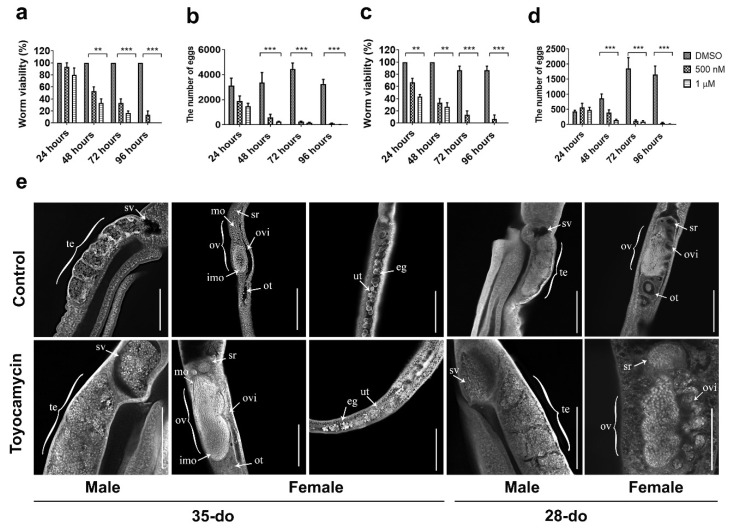
The effect of toyocamycin on viability and egg production of 35- or 28-do adult worms of *Schistosoma japonicum* and CLSM (confocal laser scanning microscopy) study of gonads of 35- or 28-do adult worms after toyocamycin treatment. (**a**) Effect of toyocamycin treatment between 24 and 96 h on viability of 35-do worms. (**b**) Effect of toyocamycin treatment between 24 and 96 h on egg production in 35-do worms. (**c**) Effect of toyocamycin treatment between 24 and 96 h on viability in 28-do worms. (**d**) Effect of toyocamycin treatment between 24 and 96 h on egg production in 28-do worms. (**e**) CLSM analyses of paired worms (35- or 28-do) incubated with DMSO (control) or 1 µM toyocamycin for 96 h. Abbreviations: te, testes; sv, sperm vesicle; ov, ovary; mo, mature oocytes; imo, immature oocytes; ovi, oviduct; ot, ootype; ut, uterus; sr, seminal receptacle; eg, egg. Scale bars: 200 µm. Data are representatives of the mean ± SD of three independent experiments. ** indicates *p* < 0.01; *** indicates *p* < 0.001.

**Table 1 pathogens-10-00862-t001:** Accession numbers of RIOK-1 genes used for the multiple alignments and phylogeny in Figure 1. The RIOK-1 of the yeast *Saccharomyces cerevisiae* (CAA99317.1) was used as outgroup.

Species	Accession Numbers	References
*Saccharomyces cerevisiae* ^**1**^	CAA99317.1	[47]
*Xenopus laevis*	NP_001116165.1	[48]
*Xenopus tropicalis*	XP_004915351.1	[49]
*Homo sapiens* ^**2**^	NP_113668.2	[50]
*Archaeoglobus fulgidus* ^**2**^	NP_73535983	[51]
*Arabidopsis thaliana* ^**2**^	NP_180071.1	[52]
*Danio rerio* ^**2**^	NP_998160.1	[53]
*Rattus norvegicus*	NP_001092981.1	[54]
*Mus musculus*	NP_077204.2	[55]
*Oryza sativa*	BAC79649.1	[56]
*Arabidopsis thaliana* ^**2**^	NP_180071.1	[52]
*Drosophila melanogaster*	NP_648489.1	[57]
*Ascaris suum*	ERG87084.1	[58]
*Trichostrongylus vitrinus*	CAR64255.1	[59]
*Haemonchus contortus* ^**2**^	ADW23592.1	[6]
*Caenorhabditis elegans* ^**2**^	CCD67367.1	[8]
*Clonorchis sinensis* ^**2**^	GAA42679.2	[60]
*Echinococcus granulosus* ^**2**^	EUB62820.1	[61]
*Schistosoma haematobium* ^**2**^	XP_012792680.1	[42]
*Schistosoma mansoni* ^**2**^	CCD59229.1	[62]

**^1^** Sequence used as an outgroup in phylogenetic analysis. **^2^** Sequence used in the alignment.

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
