# Peer review of "First Evidence of Function for *Schistosoma japonicum"

_pathogens, 2021, doi:10.3390/pathogens10070862_

Round 1

Reviewer 1 Report

Mudassar N. Mughal et al identified and characterized gene riok-1 from S. japonicum and showed its important function in regulating the developmental and/or reproductive processes in S. japonicum by using RNAi and chemical inhibition treatments. This study is interesting, however, I have several concerns about this study:

  1. It will be good if authors would provide more information about the relationship or difference between riok-1 and riok-2? Including homology between then in schistosome.
  2. In the results section 2.2, authors showed that “the transcript level of Sj-riok-1 in single, cultured females for 3 d was 7.2-fold higher than in couples cultured for the same time”. As we understood, pairing is important for female schistosome reproductive development, would you please explain why the riok-1 expression level was increased in single females compared with paired female S. japonicum? Have you checked the riok-1 transcription level in single male worms?
  3. In the toyocamycin treatment experiment, have you tested the transcription of riok-1 in the inhibitor treated worms? CLSM examination was used to check the oocytes, it will be good to use this method to measure size of ovary to further support the results obtained in RNAi.
  4. Fig4d, the concentration of toyocamycin should be 1µM, not 1M
  5. In the Materials and Methods section 4.5, authors descripted that “Both 28-do and 35-do couples were treated 288 with varying concentrations (100 nM, 300 nM, 500 nM and 1 μM) of toyocamycin in vitro 289 for 96 h.” It will be good to show all the results of worm viability and egg numbers after being treated with different concentrations of inhibitor, by providing the IC500 or other information.

Author Response

We are grateful for the valuable suggestions, which were very helpful indeed.

Reviewer 2 Report

This manuscript describes an investigation into the biological role for RIOK-1 in the development of S. mansoni parasites. It revealed evidence of heightened expression of this kinase in the schistosomula and female compared to the other life stages. With a focus on the female worms, the study then progressed to show that inhibition of riok-1 expression by siRNA, of inhibition of kinase activity by administration of toyocamycin, impacted the development of ovaries and significantly reduced the numbers of eggs produced. As such this study has uncovered a possible new target for future anti-helminthic drug development which is very timely given the ever growing resistance to PZQ and lack of alternative options.

The experimental approach followed the standard pathway of genome analysis, expression studies, knockdown/inhibition with all studies performed well with appropriate critical analysis.

I have some minor comments:

  1. In the phylogenetic analysis, it would be useful to include the Fasciola worms - as these also inhabit the liver, and commonly use as a comparison for Schistosome parasite, I was surprised that they were absent from Fig 1.
  2. Fig 2 - while the quantification of RNA expression is explained in the methods, it would be useful if there was some reference to what the expression is "relative" to in either the legend or on the axes of the graphs here
  3. Line 127: I would suggest that this sentence is tempered somewhat - looking at the data presented I would suggest that there is likely no significant difference between the levels of expression in the schistosomula versus the female worms, so to state that the highest expression is in the adult stage is not quite an accurate representation of the data.
  4. Can the authors clarify (either in the methods or the legend) whether the female worms examined in Fig 2a, are from mixed or single cultures.
  5. Line 145; I would argue that Toyocamycin is not "specific" for RIOK - on a quick search it would seem to have broader inhibitory action that just the RIO kinases. The description here should be edited to reflect this and perhaps also addressed in the discussion - is it possible the effects you are seeing are due to other inhibitory actions?
  6. Fig 4a-d; I assume the drug dose listed in the graphs should be 1uM

Author Response

We are grateful for the valuable suggestions and constructive input, which were very helpful to improve the clarity to readers.
